# Cross-Class Domain Adaptive Semantic Segmentation with Visual Language Models

## ABSTRACT

This paper addresses the issue of cross-class domain adaptation (CCDA) in semantic segmentation, where the target domain contains both shared and novel classes that are either unlabeled or unseen in the source domain. This problem is challenging, as the absence of labels for novel classes hampers the accurate segmentation of both shared and novel classes. Since Visual Language Models (VLMs) are capable of generating zero-shot predictions without requiring task-specific training examples, we propose a label alignment method by leveraging VLMs to relabel pseudo labels for novel classes. Considering that VLMs typically provide only image-level predictions, we embed a two-stage method to enable fine-grained semantic segmentation and design a threshold based on the uncertainty of pseudo labels to exclude noisy VLM predictions. To further augment the supervision of novel classes, we devise memory banks with an adaptive update scheme to effectively manage accurate VLM predictions, which are then resampled to increase the sampling probability of novel classes. Through comprehensive experiments, we demonstrate the effectiveness and versatility of our proposed method across various CCDA scenarios.

## CCS CONCEPTS

• **Computing methodologies → Image segmentation**.

## KEYWORDS

Cross-class domain adaptation, Semantic segmentation, Visual language models

## 1 INTRODUCTION

Traditional Unsupervised Domain Adaptation (UDA) methods for semantic segmentation [12, 24, 42, 49] aim to transfer knowledge from a labeled source domain (e.g. synthetic samples) to an unlabeled target domain (e.g. real-world samples). These methods, while typically relying on self-training with pseudo-label generation [11, 13, 35, 47] for better adaptation, have become a promising solution for addressing cross-domain challenges. However, due to the assumption that the source and target domains must share an identical set of classes, existing UDA solutions suffer from limited generalizability especially when confronted with previously unseen classes in the target domain [20, 25, 27]. This limitation renders

*ACM MM, 2024, Melbourne, Australia*
© 2024 Copyright held by the owner/author(s). Publication rights licensed to ACM.
ACM ISBN 978-x-xxxx-xxxx-x/YY/MM
https://doi.org/10.1145/nnnnnnn.nnnnnnn

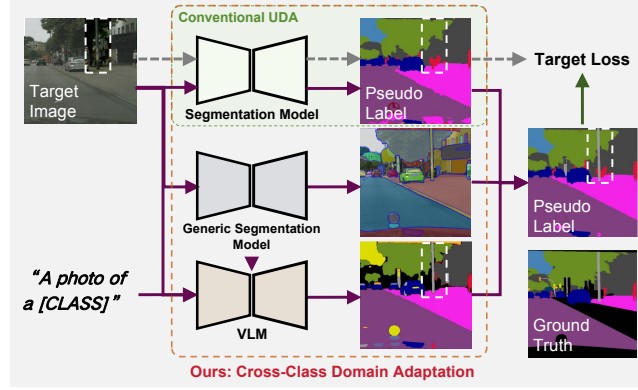

**Figure 1: Comparison between previous UDA methods and our method. Here, the novel class is defined as "*pole*", and the ground truth is provided for better visualization. When encountering the novel class, conventional UDA methods like [11] fail to capture correct pseudo labels, where the novel class is always predicted as the shared ones. Instead, by utilizing VFMs [7, 16, 28], our model can relabel novel classes with accurate annotations, which facilitates robust learning of both shared and novel classes.**

them inflexible in real-world scenarios, where encountering unseen classes is a common occurrence.

Motivated by this, we focus on a more practical but challenging problem named *Cross-Class Domain Adaptation* (CCDA) [8] for semantic segmentation, where the target domain not only includes the *shared classes* but also involves some private *novel classes* that are unlabeled or unseen in the source domain. TACS [8] is a pioneer in researching this issue, but they require 30 labeled samples for each novel class. Different from TACS, our objective is to address challenges across different classes and domains *without relying on additional labels for novel classes*. However, this setting is extremely challenging as the absence of novel class labels hampers the correct segmentation of both shared and novel classes. Specifically, the lack of novel class annotations decreases the reliability of pseudo labels by assigning novel classes as shared ones (as shown in Figure 1), causing more noisy pseudo labels to mislead the training in the target domain.

Recent breakthroughs in contrastive pre-training studies present a promising path for developing foundational models that integrate vision and language, known as Vision-Language Models (VLMs), for various computer vision tasks. These VLMs like CLIP [28] and ImageBind [7] excel at encoding a diverse range of visual concepts and demonstrate remarkable capabilities in making zero-shot predictions without task-specific training examples [34, 41], presenting a potential solution to segment novel classes in CCDA. Nevertheless, VLMs are trained to match an entire image with a text description

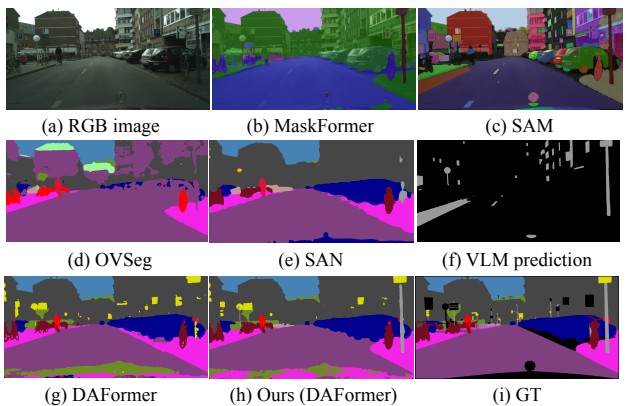

(a) RGB image    (b) MaskFormer    (c) SAM

(d) OVSeg    (e) SAN    (f) VLM prediction

(g) DAFormer    (h) Ours (DAFormer)    (i) GT

**Figure 2: Qualitative visualization. (a) is the input image. (b) and (c) depict widely used mask proposal generators that predict class-agnostic segmentation maps. However, Mask-Former [3] fails to capture precise segmentation maps, while SAM [16] can not provide masks for each pixel. (d) and (e) are existing SOTA works, OVSeg [19] and SAN [44], performing fine-grained semantic segmentation with VLMs, where they exhibit low performance on small-size objects and distort the outlines of objects. (f) denotes the mask proposals of the novel class "*pole*" produced by combining SAM [16] and ImageBind [7]. (g) and (h) are the results generated by the DAFormer [11] framework without and with our proposed elements. The ground truth is shown in (i).**

and fail to capture fine-grained alignment between image regions and text, resulting in limited applicability in fine-grained semantic segmentation tasks. A promising remedy for enabling fine-grained predictions is to generate a set of class-agnostic mask proposals (i.e., Figure 2 (b-c)), and then leverage VLMs to classify them into specific classes [19, 44, 45]. For example, simply combining SAM [16] and ImageBind [7] can generate Figure 2 (f) for the novel class "pole". However, these VLM-based fine-grained segmentation methods demonstrate limited performance (as shown in Figure 2 (d-f)) due to several factors: (1) the visual disparities between VLMs' training images and masked proposals [19, 45], (2) the defective predictions of the mask proposal generators. Thus, CCDA challenges are far from solved by directly applying these VLM-based approaches.

To bridge the research gap, in this paper, we present a label alignment approach for novel classes by employing VLMs to relabel pseudo labels. In detail, we propose to utilize VLMs for locating novel classes within target images, and then relabel pseudo labels with these novel-class proposals. Motivated by evident failure cases where several shared-class regions are misclassified as novel ones in VLM predictions (i.e., Figure (f)), we present an uncertainty-driven adaptive threshold for excluding misclassified proposals and alternatively performing label alignment. To further enable the sufficient learning of novel classes, we establish memory banks to store novel-class proposals and resample them to augment pseudo labels. Given the limited memory size, it is neither feasible nor necessary to memorize every novel class proposal. To address this issue, we devise an adaptive memory update scheme to dynamically

incorporate new proposals while discarding outdated ones. We comprehensively assess the effectiveness and universality of our method across different benchmarks, UDA frameworks, and VLM-based segmentation methods. Extensive experiments illustrate our method surpasses the existing state-of-the-art (SOTA) approaches under the CCDA setting (as depicted in Figure 2 (g-i)). In summary, our **contributions** can be outlined as follows:

- We propose a VLM-based label alignment method with an uncertainty-driven threshold to relabel pseudo labels with accurate novel class proposals.
- We present a novel class resampling method with adaptively updated memory banks to augment pseudo labels, which facilitates the sampling probability of novel classes.
- Extensive evaluations validate that our proposed model achieves top performance on different datasets and scenarios in terms of both effectiveness and universality.

## 2 RELATED WORK

### 2.1 Unsupervised domain adaptation

UDA semantic segmentation has been maturely developed recently, which can be divided into adversarial training methods and self-training methods. Adversarial training approaches struggle to bridge the domain gap by aligning the distributions of source and target domains in either the input level [10, 33, 48], feature level [5, 22], output level [30, 39, 46], or patch level [37] based on a Generative Adversarial Network (GAN) [9]. Benefiting from their training stability and strengthened performance, self-training methods, which strive to generate pseudo labels offline [32, 51] or online [6, 11, 35, 50] for enabling the training of the target domain, gradually become the main solution for UDA. Moreover, some work [15, 40] also attempts to combine adversarial training with self-training strategies, pursuing better performance in cross-domain scenarios. However, these UDA methods assume that the source and target domains must possess identical sets of classes, rendering them vulnerable when confronted with novel classes [20, 25, 27]. This limitation hinders their applications in real-world scenarios. Therefore, investigating the CCDA problem is highly necessary for semantic segmentation tasks.

### 2.2 Cross-class domain adaptation

Driven by the restriction of UDA methods on encountering novel classes, some recent studies [18, 27, 31] attempt to explore the cross-class problem in domain adaptation scenarios, where the target domain shares some classes with the source domain, but it also includes its own distinct, private novel classes. For instance, open-set/universal domain adaptation works [27, 31] focus on assigning the "unknown" label to all novel classes during the inference, while class-incremental domain adaptation [18] aims at recognizing each shared and novel target classes with a unique semantic label. Nevertheless, these studies merely revolve around image classification tasks, resulting in CCDA for semantic segmentation still being an open question. TACS [8] is the work most similar to ours. However, TACS still necessitates few-shot labels for novel classes to conduct cross-class adaptation, which restricts its generalization capability in more challenging zero-shot scenarios. In contrast to TACS

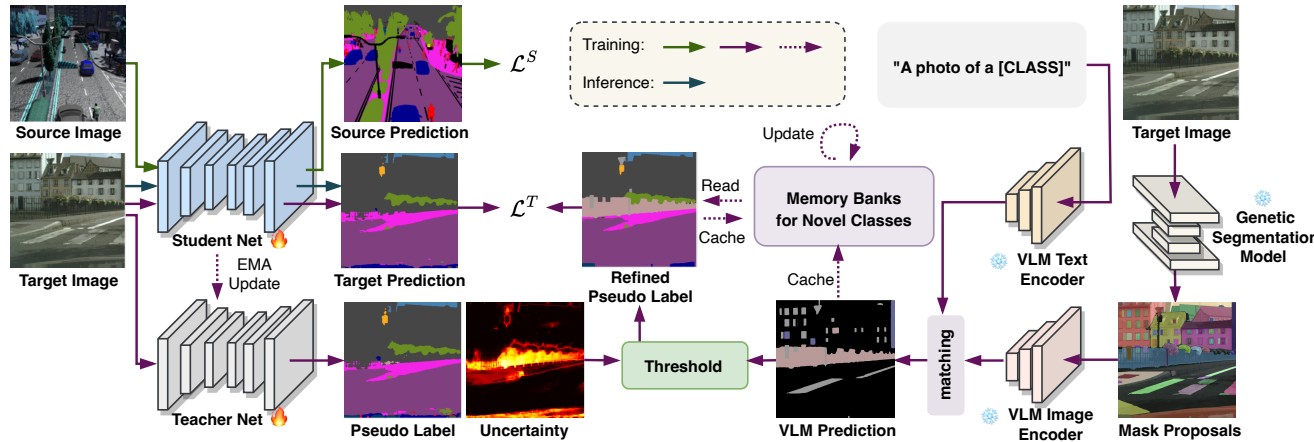

**Figure 3: Overview of our framework, where novel classes are wall, fence, pole. Due to the lack of novel class labels, the segmentation model predicts novel classes as shared ones. In this work, we propose a VLM-based label alignment method to relabel pseudo labels with accurate novel class proposals and present a novel class resampling method with memory banks to further augment the supervision of novel classes.**

[8], we attempt to address CCDA challenges without relying on additional labels for novel classes.

### 2.3 Pre-trained visual-language models

Due to the advancements in self-supervised representation learning [1, 2], multimodal Visual Language Models (VLMs), designed to acquire generic representations of vision and language, have showcased remarkable achievements in recent years. Recent studies, exemplified by CLIP [28] and ImageBind [7], connect visual and language concepts by conducting pre-training on large-scale text-image pairs. As numerous vision and language concepts are encompassed within large-scale datasets, these methods showcase surprisingly robust capabilities in conducting zero-shot predictions, marking significant progress in various downstream tasks. For example, Simseg [45] proposes a two-stage segmentation framework, where they first generate class-agnostic mask proposals within images and then assign specific semantics for each mask by a frozen CLIP. SAN [44] presents an adapter network to benefit CLIP in recognizing the semantics of mask proposals. However, despite their successes, the performance of these VLM-based segmentation methods [19, 44, 45] remains constrained in complex scenarios due to suboptimal mask proposal generators [3, 16] and visual disparities between training and test samples [19]. In this paper, we endeavor to harness the capabilities of VLMs for relabeling pseudo labels, offering a promising avenue for their application in driving scenes.

## 3 METHODOLOGY

### 3.1 Problem description

In the CCDA settings, we have a labeled source dataset $\{\mathcal{X}, \mathcal{Y}\}^S = \{(x_n^s, y_n^s)\}_{n=1}^{N^S}$ and an unlabeled target dataset $\{\mathcal{X}^T\} = \{x_n^t\}_{n=1}^{N^T}$, where $\mathcal{X}$ and $\mathcal{Y}$ refer to images and semantic labels. In the source dataset, each image $x_n^s$ is associated with a corresponding label $y_n^s$ of source classes $C_{shared}$. However, in the target dataset, in addition to $C_{shared}$, there are several private novel classes $C_{novel}$

($C_{shared} \cap C_{novel} = \varnothing$) that are either unlabeled or unseen in the source dataset. In this context, the label space of the source domain $C^S = C_{shared}$, and that of the target domain $C^T = C_{shared} \cup C_{novel}$. Our objective is to adapt the segmentation model $f_\theta$, trained on $\{\mathcal{X}, \mathcal{Y}\}^S$ and $\mathcal{X}^T$, to target dataset samples with the label space $C^T$.

### 3.2 Framework overview

In our framework, as shown in Figure 3, a neural network $f_\theta$ is trained on the source dataset $(\mathcal{X}^S, \mathcal{Y}^S)$ and is expected to perform well on the target dataset samples without access to target labels $\mathcal{Y}^T$. As annotations $\mathcal{Y}^S$ are available for the source classes $C^S$, we naively compute the supervised cross-entropy loss $\mathcal{L}^S$ on the source domain via:

$$\mathcal{L}^S = -\sum_{i=1}^{H}\sum_{j=1}^{W}\sum_{c=1}^{C^S} y_{i,j,c}^s \log f_\theta(x^s)_{i,j,c}. \tag{1}$$

Nevertheless, the model only trained with $\mathcal{L}^S$ will suffer from performance degradation on the target domain. To address this issue, an auxiliary unsupervised loss $\mathcal{L}^T$ is devised for the target domain, which is calculated with pseudo labels $\hat{y}^t$ predicted via a teacher network $f_{\theta'}$:

$$\hat{y}_{i,j,c}^t = \xi(c = \arg\max_{c' \in C^T} f_{\theta'}(x_{i,j,c'}^t)). \tag{2}$$

$$\mathcal{L}^T = -\sum_{i=1}^{H}\sum_{j=1}^{W}\sum_{c=1}^{C^T} q^t \hat{y}_{i,j,c}^t \log_\theta(x^t)_{i,j,c}. \tag{3}$$

Here, $\xi(\cdot)$ and $q^t$ represent the Iverson bracket and the confidence weighting coefficient [12, 13], respectively. Then the total loss $\mathcal{L}$ is computed as $\mathcal{L} = \mathcal{L}^S + \mathcal{L}^T$. In addition, to stabilize pseudo labels [11], the teacher network $f_{\theta'}$ is optimized via the exponential moving average (EMA) strategy [17] by:

$$\theta' \leftarrow \alpha * \theta' + (1 - \alpha) * \theta. \tag{4}$$

Here, $\alpha$ is the smoothing factor and is defined as 0.999.

However, due to the absence of novel class labels, the segmentation model struggles to capture robust representations for novel classes during training, resulting in novel classes being erroneously assigned as shared ones in pseudo labels. Driven by this motivation, we first present a VLM-guided label alignment method to alternatively rectify pseudo labels with precise proposals for novel classes. Additionally, to enhance sufficient learning for novel classes, we construct memory banks to dynamically manage novel class proposals and utilize them to augment the supervision for novel classes. Detailed descriptions of our proposed methods for label alignment and novel class resampling will be provided in the following sections.

### 3.3 Label alignment

Under the CCDA settings, the segmentation model consistently develops a biased perception of novel classes due to the absence of their annotations. Hence, we propose a label alignment method aimed at rectifying pseudo labels with accurate novel class semantics. Our label alignment method comprises two steps: mask proposal generation for novel classes and threshold-guided pseudo label rectification.

**Mask proposal generation for novel classes.** The correction of pseudo labels necessitates precise semantic maps for novel classes. However, given unlabeled target samples, localizing novel classes with accurate semantics poses a significant challenge. To address this issue, we resort to pre-trained VLMs, especially ImageBind [7], thanks to their capability of making zero-shot predictions without the need for task-specific training examples. Specifically, in this work, we introduce a VLM-based fine-grained segmentation method to conduct the localization of novel classes: we first generate class-agnostic mask proposals from target images and then assign each mask proposal with a specific semantic.

In particular, we segment objects within an unlabeled target image $x^t$ based on a generic segmentation model $g_{seg}$, where the process can be formulated as follows:

$$\mathcal{M} = g_{seg}(x^t), \tag{5}$$

Here, $\mathcal{M}$ represents a set of binary masks. Note that each mask covers a potential object in $x^t$ but fails to deliver any semantic labels. By using each mask $m \in \mathcal{M}$, we extract an individual object from the color image and capture its unique mask proposal via $w = m \odot x^t$.

We then classify these mask proposals with the guidance of ImageBind. The fundamental concept is to introduce the text encoder $\phi_{text}$ and the image encoder $\phi_{img}$ of ImageBind to conduct semantic alignment between visual proposals and text descriptions, identifying novel classes in unlabeled target samples.

Specifically, let "a photo of a [CLASS]", denoted as $\mathcal{A}$, refer to the semantic template that serves as input to $\phi_{text}$. This input produce a $d$-dimensional text embedding $e_c$ for the class $c$ via:

$$e_c = \phi_{text}(\mathcal{A}), e_c \in \mathbb{R}^{1 \times d}. \tag{6}$$

In Eq. (6), we obtain a set of text embeddings for the classes in $C^T$. Similarly, for each mask proposal $w$, we can derive a visual embedding $u$ by $\phi_{img}$:

$$u = \phi_{img}(w), u \in \mathbb{R}^{1 \times d}. \tag{7}$$

We calculate the cosine distance between $u$ and the set of text embeddings. Then, the most similar text embedding is selected via:

$$c^* = \arg\max_{c \in C^T} \frac{u \cdot e_c}{||u|| \cdot ||e_c||}. \tag{8}$$

Here, $|| \cdot ||$ represents L2-norms. The class semantic of mask proposals $w$ is assigned as $c^*$ if $c^* \in C_{novel}$, where we denote this novel class proposal as $w^*$. Hence, we generate a set of novel class proposals by repeating the aforementioned process on each unlabeled target training image $x^t$. Nevertheless, due to the visual disparities between masked images and ImageBind's training samples, there are numerous noisy predictions in labeled proposals, where a considerable number of shared class proposals are misclassified as novel ones (as shown in Figure. 3).

**Threshold-guided pseudo label rectification.** To tackle the problem of noisy VLM predictions, we design an uncertainty-driven adaptive threshold to filter out misclassified proposals and conduct label alignment with accurate ones. Our proposed method is derived from the empirical observation that novel classes exhibit significant uncertainty compared with shared ones by the segmentation model [14, 21, 43]. Therefore, the uncertainty of shared classes can be regarded as a threshold for localizing novel class regions. Specifically, given a novel class proposal $w^*$ and its associated image region $w$, we regard $w^*$ as a misclassification if the segmentation model yields higher uncertainty in $w$. In this paper, we adopt the entropy $v$ of the pseudo label $\hat{y}^t$ as the prediction uncertainty, which is defined as:

$$v_{i,j} = -\frac{1}{\log(C^T)} \sum_{c=1}^{C^T} p_{i,j,c} \log p_{i,j,c}. \tag{9}$$

Here, $p_{i,j,c}$ represents the softmax probability of pixel $x_{i,j}$ in class $c$. However, due to varying entropy exhibited throughout the self-training process, and the inconsistent complexities among different classes, the prediction entropy is both class-dependent and time-varying. Hence, it is impractical to compare the entropy of novel and shared classes. Aiming at this challenge, to capture real-time entropy at different training stages, we calculate the average entropy $\bar{v}_c$ for each class $c \in C^T$ at every iteration step and then employ the EMA strategy for their online updates via:

$$\bar{v}_c = \frac{\sum_{i=1}^{H} \sum_{j=1}^{W} \delta(\hat{y}_{i,j}^t == c) * v_{i,j}}{\sum_{i=1}^{H} \sum_{j=1}^{W} \delta(\hat{y}_{i,j}^t == c)}, \tag{10}$$

$$\bar{v}_c \leftarrow \beta * \bar{v}_c' + (1 - \beta) * \bar{v}_c. \tag{11}$$

Here, $\bar{v}_c'$ is the mean entropy of the class $c$ in the previous iteration, and $\beta$ is the smoothing factor set as 0.999.

Considering a pixel $x_{i,j}$ located in a proposal $w^*$ labeled with $c_{novel}$, the segmentation model predicts $x_{i,j}$ as $c_{shared}$ with entropy $v_{i,j}$. It is intuitive to conclude that the segmentation model makes an inaccurate prediction at $x_{i,j}$ when $v_{i,j} > \bar{v}_{c_{shared}}$, and an accurate prediction when $v_{i,j} \leq \bar{v}_{c_{shared}}$. Inspired by this, we relabel $\hat{y}^t$ via:

$$\hat{y}_{i,j}^t = \begin{cases} c_{novel}, & v_{i,j} > \sigma_1 * \bar{v}_{c_{shared}}, \\ c_{shared}, & otherwise. \end{cases} \tag{12}$$

Here, the scaling factor $\sigma_1$ is introduced to exclude cases where the object is misjudged due to some factors such as distance or size.

Nevertheless, Eq. (12) is executed discretely in spatial dimensions, ignoring the challenging regions on shared classes and hindering the generation of complete semantic maps for novel classes. Therefore, we further employ a voting strategy to address these issues in pseudo-label rectification. This voting strategy is conducted within the area covered by $m$ in $\hat{y}^t$, utilizing statistical analysis of pixel-wise semantics:

$$z_c = \frac{\sum\limits_{i=1}^{H} \sum\limits_{j=1}^{W} \delta(\hat{y}_{i,j}^t == c) * \delta(m_{i,j} == 1)}{\sum\limits_{i=1}^{H} \sum\limits_{j=1}^{W} \delta(m_{i,j} == 1)}, \tag{13}$$

$$c^* = \arg\max_{c \in C^T} z_c. \tag{14}$$

Here, $\delta(\cdot)$ is the indicator function. The pseudo label $\hat{y}^t$ is further refined via:

$$\hat{y}^t \leftarrow m \odot (\mathcal{I} * c^*) + \bar{m} \odot \hat{y}^t. \tag{15}$$

$\mathcal{I}$ denotes a matrix filled with ones, which shares the same shape with $\hat{y}^t$.

## 3.4 Novel class resampling

Although the label alignment method is capable of rectifying pseudo labels, the quantity of corrected novel class regions remains insufficient to enable comprehensive learning by the segmentation model. Hence, we establish memory banks to store novel class proposals for further enhancing the supervision of novel classes. However, storing all novel class proposals generated by VLMs is impractical, as it would accumulate noisy proposals and significantly inflate the size of the memory bank. Therefore, we propose a fixed-size memory bank with an adaptive update strategy to more effectively manage novel class proposals. Specifically, our proposed adaptive update strategy comprises two operations: caching new proposals and removing obsolete ones.

**Caching new proposals.** Since there is a considerable number of misclassifications in VLM predictions, simply storing these novel class proposals generated offline may involve misleading information. Therefore, we maintain an independent memory bank for each novel class online for organizing those proposals with accurate labels. In this work, the proposals intended for storage primarily originate from three sources. Firstly, executing the operations outlined in Eq. (5-15) yields a comprehensive appearance of the novel class region. These proposals, after being rectified by novel class alignment, are stored in the memory bank. Secondly, in the later stages of self-training, our label alignment method enables the segmentation model to accurately represent novel class objects. We incorporate these novel class regions, self-predicted by the segmentation model, into our memory bank for subsequent pseudo-label augmentation. Thirdly, to accommodate a more diverse range of novel class proposals, we relax the requirement in Eq. (14) that the novel class must constitute the majority. Specifically, given a VLM-predicted proposal $w^*$ labeled with $c_{novel}$ and $\hat{y}^t$ relabeled after label alignment, we calculate the ratio $z_{c_{novel}}$ via Eq.(14). Once $z_{c_{novel}} > \sigma_2$, where $\sigma_2$ is a predetermined threshold to mitigate the interference caused by challenging regions of shared classes, $w^*$ is adopted for storage.

**Removing obsolete proposals.** In this study, we eliminate obsolete proposals based on the pixel areas of cached samples, as larger-area proposals contribute significantly to the comprehensive learning of novel classes [36]. Instead of directly counting the pixel areas of cached samples, we preserve valuable proposals based on the number of pixels $p$ predicted or rectified as novel classes, as it indirectly reflects both the pixel area and discriminative capacity of a specific proposal. Specifically, within a memory bank of $c_{novel}$, we preserve the subset $D_{c_{novel}}$ consisting of $K$ proposals with the highest $p$ values. Furthermore, we utilize these cached novel class proposals to augment pseudo labels via ClassMix [26]:

$$\hat{y}^t \leftarrow m \odot w^* + \bar{m} \odot \hat{y}^t. \tag{16}$$

# 4 EXPERIMENTS

## 4.1 Experimental settings

**Datasets.** To validate the effectiveness of our proposed elements, we evaluate our framework on two benchmarks. Namely SYN-THIA → Cityscapes and SYNTHIA → IDD, respectively. The source dataset in both settings is SYNTHIA [29], where only the objects of classes *road*, *sidewalk*, *building*, *traffic light*, *traffic sign*, *vegetation*, *sky*, *person*, *rider*, *car*, *bus*, *motorcycle*, and *bike* are labeled in our experiments. In the first setting, we focus on dealing with the domain gap in structured scenes, where the Cityscapes [4] serves as the target dataset, where no annotations are given for novel classes *wall*, *fence*, *pole*, *terrain*, *truck*, and *train*. In the second setting, we attempt to address domain gaps in unstructured scenes. We leverage IDD dataset [38] as the target dataset, where 6 classes *wall*, *fence*, *pole*, *truck*, *animal*, and *autorickshaw* are defined as novel classes with no annotations.

**Implementation details.** In our experiments, the implementation is based on DAFormer [11] framework with its training hyperparameters and self-training strategy. Our model is trained with AdamW [23], where a learning rate of $6 \times 10^{-5}$ is set for the encoder and that of $6 \times 10^{-4}$ for the decoder. We configure a weight decay to 0.01, and the batch size to 2, where the images are cropped into $512 \times 512$ [11] during the $40k$ iterations. We adopt SAM [16] as our generic segmentation model due to its capability on precisely segmenting arbitrary images. We fixed the size of the memory bank for each novel class at 30. When referring to the parameters involved in our elements on the DAFormer framework, we set $\sigma_1$ and $\sigma_2$ to 2 and 0.1 for best performance.

**Competitive methods.** To evaluate the effectiveness in zero-shot CCDA scenarios, our proposed model is compared with the existing top-performing algorithms, which include UDA methods [11–13, 35], the VLM-based methods [7, 16, 19, 44], and the CCDA methods [8].

## 4.2 Comparison with SOTA methods

In this section, we evaluate our method against other top-performing methods in various cross-class scenarios. Tables 1 and 2 illustrates the segmentation results on both benchmarks.

**Comparison with UDA methods.** We first assess our proposed method based on DAFormer [11] framework with SOTA UDA methods [11–13, 35]. As shown in Table 1 and 2, our proposed method achieves top performance and outperforms these UDA methods.

**Table 1: Quantitative comparison (IoU in %) with the SOTA methods on the SYNTHIA → Cityscapes benchmark. The results of novel classes are emphasized in gray. The best result is highlighted in bold.**

| | Method | Road | S.walk | Build. | Wall | Fence | Pole | Tr.L. | Tr.S. | Veg. | Terr. | Sky | Person | Rider | Car | Truck | Bus | Train | M.bike | Bike | MIoU | MIoU |
|---|---|---|---|---|---|---|---|---|---|---|---|---|---|---|---|---|---|---|---|---|---|---|
| UDA | DACS [35] | 82.97 | 24.47 | 79.77 | 0.00 | 0.00 | 0.00 | 51.97 | 36.47 | 79.96 | 0.00 | 90.38 | 71.89 | 33.45 | 87.85 | 0.00 | 35.40 | 0.00 | 32.92 | 59.27 | 0.00 | 40.35 |
| | DAFormer [11] | 86.84 | 46.79 | 84.89 | 0.00 | 0.00 | 0.00 | 52.32 | 53.17 | 83.24 | 0.00 | 83.87 | 70.44 | 43.57 | 87.56 | 0.00 | 51.70 | 0.00 | 53.69 | 62.43 | 0.00 | 45.29 |
| | HRDA [12] | 80.64 | 51.53 | 83.62 | 0.00 | 0.00 | 0.00 | **64.64** | 65.17 | 80.12 | 0.00 | 92.98 | 64.98 | 54.65 | 87.55 | 0.00 | 21.49 | 0.00 | 62.23 | 64.72 | 0.00 | 46.02 |
| | MIC [13] | 87.76 | **56.72** | 85.78 | 0.00 | 0.00 | 0.00 | 63.50 | **70.14** | 84.50 | 0.00 | 93.00 | 80.50 | **58.98** | 83.78 | 0.00 | 62.81 | 0.00 | **65.27** | 65.88 | 0.00 | 50.45 |
| VLM | S+I [7, 16] | 51.60 | 10.62 | 39.47 | 12.74 | 16.08 | 4.91 | 3.17 | 12.46 | 39.79 | 10.18 | 21.84 | 5.08 | 0.36 | 31.52 | 19.55 | 25.16 | 4.69 | 10.66 | 21.04 | 11.36 | 17.94 |
| | OVSeg [19] | 53.21 | 20.46 | 52.08 | 15.77 | 21.67 | 2.13 | 23.10 | 20.51 | 31.30 | 1.32 | 73.85 | 56.31 | 10.23 | 53.23 | 34.56 | 21.20 | 0.38 | 30.69 | 46.01 | 12.64 | 29.90 |
| | SAN [44] | 85.03 | 44.67 | 78.48 | 33.48 | **41.79** | 2.97 | 37.39 | 26.45 | 78.49 | 0.00 | 80.96 | 58.03 | 0.00 | 43.29 | 23.05 | 58.00 | 0.68 | 30.64 | 55.61 | 17.00 | 41.00 |
| CCDA | TACS [8] | 72.47 | 27.23 | 81.06 | 0.00 | 0.00 | 0.00 | 29.40 | 25.86 | 76.93 | 0.00 | 89.84 | 68.14 | 36.33 | 84.04 | 0.00 | 40.59 | 0.00 | 36.25 | 43.11 | 0.00 | 37.43 |
| | Ours (DACS) | **89.48** | 39.04 | 81.77 | 7.67 | 29.32 | **28.75** | 52.37 | 44.35 | 86.16 | 22.85 | 81.19 | 71.47 | 30.49 | 87.75 | 19.53 | 23.56 | 26.10 | 36.77 | 65.18 | 22.37 | 48.51 |
| | Ours (DAFormer) | 84.06 | 39.66 | **89.02** | **35.55** | 38.59 | 26.43 | 52.50 | 59.99 | **87.00** | **40.31** | 93.82 | 73.96 | 46.06 | 91.68 | **63.83** | 69.07 | **45.28** | 55.71 | 68.41 | **41.67** | **61.10** |
| | Ours (MIC) | 84.93 | 40.43 | 88.99 | 32.69 | 34.72 | 23.99 | 58.51 | 59.89 | 80.08 | 31.63 | **94.17** | **77.11** | 49.42 | **92.10** | 56.44 | **69.62** | 42.59 | 62.19 | **73.31** | 37.01 | 60.67 |

**Table 2: Quantitative comparison (IoU in %) with the SOTA methods on SYNTHIA → IDD benchmarks.**

| | Method | Road | S.walk | Build. | Wall | Fence | Pole | Tr.L. | Tr.S. | Veg. | Sky | Person | Rider | Car | Truck | Bus | M.bike | Bike | Animal | Auto. | MIoU | MIoU |
|---|---|---|---|---|---|---|---|---|---|---|---|---|---|---|---|---|---|---|---|---|---|---|
| UDA | DACS [35] | 90.26 | 6.29 | 63.43 | 0.00 | 0.00 | 0.00 | 3.22 | 7.66 | 84.49 | 92.56 | 58.81 | 53.42 | 50.39 | 0.00 | 39.09 | 53.72 | 25.89 | 0.00 | 0.00 | 0.00 | 33.12 |
| | DAFormer [11] | **95.21** | **24.95** | 67.63 | 0.00 | 0.00 | 0.00 | 13.44 | 22.11 | 86.32 | 92.80 | 66.02 | 60.99 | 47.43 | 0.00 | 42.08 | 68.97 | 24.39 | 0.00 | 0.00 | 0.00 | 37.49 |
| | HRDA [12] | 74.23 | 6.00 | 68.43 | 0.00 | 0.00 | 0.00 | 14.30 | 22.80 | 74.47 | 83.54 | 64.00 | 52.13 | 42.61 | 0.00 | 63.25 | 57.76 | 9.65 | 0.00 | 0.00 | 0.00 | 33.32 |
| | MIC [13] | 93.82 | 22.80 | 49.74 | 0.00 | 0.00 | 0.00 | **19.48** | 44.70 | 85.62 | 78.04 | 65.49 | 53.29 | 41.93 | 0.00 | 61.39 | 60.06 | 12.84 | 0.00 | 0.00 | 0.00 | 36.27 |
| VLM | S+I [7, 16] | 53.71 | 1.54 | 29.06 | 15.95 | 5.12 | 2.85 | 0.28 | 9.51 | 50.18 | 43.34 | 4.77 | 3.70 | 45.91 | 26.15 | 37.77 | 14.89 | 10.96 | 1.60 | 22.05 | 12.29 | 19.97 |
| | OVSeg [19] | 49.06 | 8.52 | 48.31 | 27.03 | 12.62 | 14.86 | 4.13 | 7.54 | 46.57 | 60.20 | 43.56 | 29.51 | 39.44 | 44.38 | 53.01 | 65.46 | 24.48 | 12.13 | 29.22 | 23.37 | 32.63 |
| | SAN [44] | 74.56 | 17.39 | 57.04 | 33.94 | 12.03 | 2.43 | 15.00 | 8.80 | 71.45 | 92.78 | 34.28 | 0.34 | 73.46 | 59.90 | 65.11 | 59.34 | 16.52 | **61.56** | 47.73 | 36.27 | 42.30 |
| CCDA | TACS [8] | 94.96 | 9.25 | 43.16 | 0.00 | 0.00 | 0.00 | 4.81 | 15.54 | 80.63 | 62.87 | 65.51 | 56.28 | 45.98 | 0.00 | 45.55 | 67.60 | 33.87 | 0.00 | 0.00 | 0.00 | 32.97 |
| | Ours (DAFormer) | 90.14 | 19.21 | **76.47** | **43.03** | **25.60** | **18.52** | 9.90 | **47.01** | **86.78** | **95.93** | **69.80** | **63.06** | **78.24** | **73.40** | **77.65** | **70.53** | **36.65** | 16.01 | **57.76** | **39.05** | **55.56** |

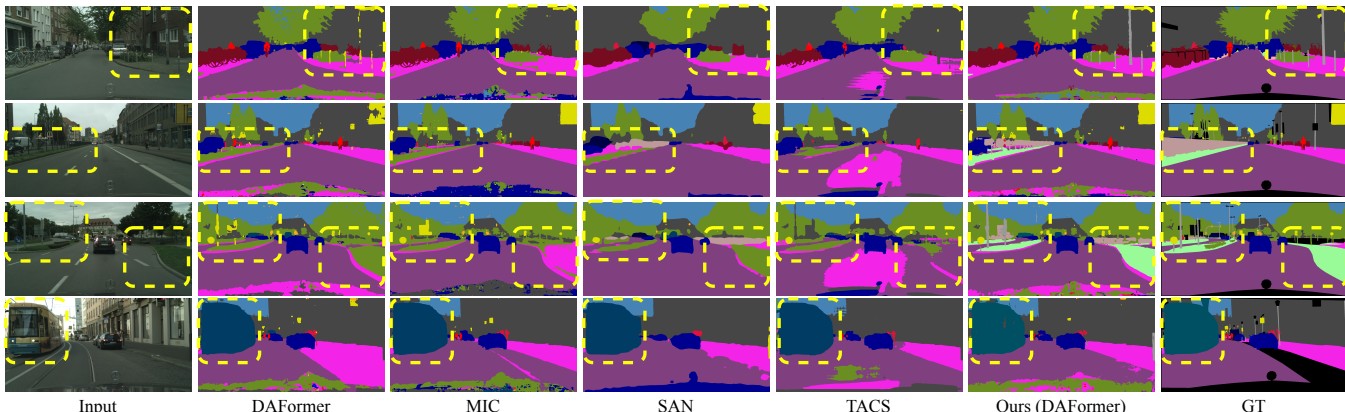

| Input | DAFormer | MIC | SAN | TACS | Ours (DAFormer) | GT |

**Figure 4: Segmentation results predicted by the previous SOTA UDA methods [11, 13], the VFM-based method [44], the TACS method [8], and our presented framework on the SYNTHIA → Cityscapes benchmark. The segmentation areas of novel classes are highlighted by yellow dashed boxes.**

Specifically, in contrast to the best performing MIC [13], our model improves by +10.65 mIoU on the SYNTHIA → Cityscapes benchmark and +17.88 mIoU on the SYNTHIA → IDD benchmark. It is worth mentioning that the performance of our method far exceeds these UDA studies on novel classes. For example, in Table

1, the novel class *pole* achieves a +26.43 significant performance gain against all UDA methods, demonstrating the effectiveness of our approach in segmenting novel classes. These observations are further supported by Figure 4, where previous methods struggle with incomplete segmentation of novel classes and misclassify

**Table 3: Ablation study of our proposed elements on the SYNTHIA → Cityscapes benchmark.**

| Method | Baseline | LA | NCR | MIoU | | |
|--------|----------|-----|-----|-------|--------|------|
| | | | | novel | shared | all |
| 1 | ✓ | | | 0.00 | 66.19 | 45.29 |
| 2 | ✓ | ✓ | | 35.99 | 69.74 | 59.08 |
| 3 | ✓ | | ✓ | 26.97 | 67.10 | 54.43 |
| 4 | ✓ | ✓ | ✓ | **41.67** | **70.07** | **61.10** |

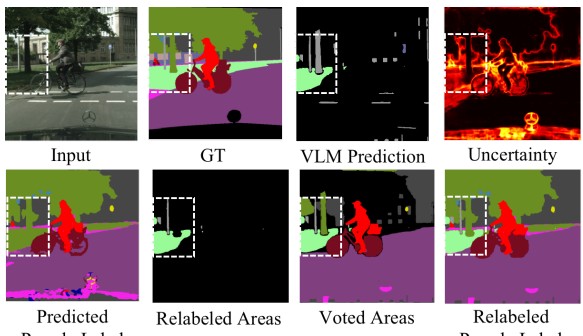

| Input | GT | VLM Prediction | Uncertainty |
|---|---|---|---|

| Predicted Pseudo Label | Relabeled Areas | Voted Areas | Relabeled Pseudo Label |
|---|---|---|---|

**Figure 5: Visualization results of pseudo labels by adopting label alignment. The segmentation results of novel classes are emphasized by white dashed boxes.**

them as shared ones. For instance, MIC misclassifies "terrain" as "vegetation" in the second row. Instead, through relabeling pseudo labels and augmenting novel classes, our model adeptly handles the challenges arising from cross-class adaptation.

**Comparison with VLM-based method.** Our method is also assessed against SOTA VLM-based fine-grained segmentation methods [7, 16, 19, 44] in Table 1, where "S+I" refers to the segmentation results generated by SAM [16] and ImageBind [7]. As exhibited in Table 1, the VLM-based methods fall short of the performance achieved by our proposed method. Specifically, the mIoU accuracy of our method stands at 61.10, surpassing the top-performing VLM method, SAN, by +20.1. In Figure 4, we observe that SAN encounters challenges in accurately segmenting tiny novel classes (i.e., "poles" in the first row), and distinguishing visually similar objects (i.e., "train" and "bus" in the fourth row). Meanwhile, the segmentation maps generated by SAN exhibit significant distortions in object contours. Therefore, there is still a lot of room for VLMs to improve fine-grained semantic segmentation results. Instead, our approach, independent of VLMs during the inference stage, achieves accurate segmentation results even in cases of small sizes or similar appearances, exhibiting superior advantages compared to VLM-based methods.

**Comparison with CCDA method.** We further evaluate our proposed method with TACS [8] which is also meticulously designed to address the CCDA problem. For comparison under our CCDA setting, we retrained TACS under the 0-shot setting, and the results obtained on the two benchmarks are shown in Tables 1 and 2. Analyzing their mIoU values, our results achieve significant performance gains compared to TACS on both benchmarks. For

instance, in Table 1, the mIoU performance of novel classes and all classes are higher than TACS by +41.67 and +23.67, respectively. The segmentation maps shown in Figure 4 also demonstrate that the elements proposed in TACS are not effective in 0-shot settings: (1) TACS fails to locate novel classes in the target images, resulting in the persisting issue of misclassifying novel classes, (2) the objects segmented by TACS exhibit incompleteness in appearance, such as the bicycles in the first row. Meanwhile, our method not only achieves correct segmentation of novel classes in a zero-shot manner but also restores the complete appearance of the objects.

**Generalization for various UDA methods.** Our proposed elements are plug-and-play and can be seamlessly embedded into self-training UDA frameworks, enabling an efficient resolution of both domain gaps and class gaps. In Tables 1 and 2, we evaluate the performance of combining our proposed elements with representative UDA methods [11, 13, 35]. It is evident that our model consistently delivers notable performance enhancements when coupled with various UDA methods. For instance, when evaluating our method with the DAFormer [11] framework on the SYNTHIA → Cityscapes benchmark, we achieve performance gains +41.67 for novel classes and +15.81 for both shared and novel classes. The performance improvements on the SYNTHIA → IDD benchmark also highlight the adaptability and versatility of our proposed elements and clarify their positive influence on various UDA methods for dealing with the CCDA problem.

## 4.3 Component ablations

To assess the effectiveness of each component, we conduct ablation studies on the DAFormer framework in Table 3 owing to the fast training. The baseline denotes the DAFormer framework retrained on CCDA settings. "LA" and "NCR" are our proposed label alignment and novel class resampling, respectively. Firstly, our LA has a positive impact on segmentation performance, resulting in a +3.55 gain in shared classes and a +35.99 in novel classes (method 1 and method 2). The absence of labels for novel classes in the baseline model impedes the robust representations of these novel classes. However, by relabeling novel classes in pseudo labels, our proposed LA method is capable of simultaneously improving the performance of both novel and shared classes. Figure 5 further illustrates the effectiveness of our presented uncertainty-guided threshold in LA. We can observe that there are a great number of misclassifications in VLM predictions. However, by comparing the uncertainty of novel classes with shared ones (i.e., the uncertainty of "terrain" and "tree"), the pseudo label can be accurately relabeled for novel class proposals. Meanwhile, the voting strategy further rectifies objects with complete masks. Secondly, the importance of NCR can not be ignored (method 1 and method 3), where mask proposals with accurate novel class labels are employed to augment pseudo labels. It plays a crucial role in facilitating the learning of novel classes while retaining the performance of shared classes. Thirdly, our complete model achieves a mIoU of 61.10, exhibiting a substantial mIoU performance improvement compared to other settings.

## 4.4 Sensitivity analysis of parameters

**Influence of the parameter** $\sigma_1$**.** To gain a more profound insight into the operational principles of our label alignment mechanism,

**Table 4: Comparison of our proposed model with different parameters on the SYNTHIA → Cityscapes benchmark.**

| | $\sigma_1$ | 0.5 | 1 | 2 | 3 | 4 |
|---|---|---|---|---|---|---|
| $\sigma_2 = 0.1$ | MIoU$_{novel}$ | 30.72 | 34.31 | **41.76** | 34.96 | 23.04 |
| | MIoU$_{all}$ | 51.15 | 55.80 | **61.10** | 58.14 | 54.14 |
| | $\sigma_2$ | 0 | 0.1 | 0.2 | 0.3 | 0.4 |
| $\sigma_1 = 2$ | MIoU$_{novel}$ | 33.86 | **41.67** | 34.77 | 33.62 | 33.73 |
| | MIoU$_{all}$ | 56.91 | **61.10** | 57.37 | 56.95 | 56.01 |

**Table 5: Segmentation results of our method with different VLM-based implements.**

| Method | MIoU$_{novel}$ | MIoU$_{shared}$ | MIoU$_{all}$ |
|---|---|---|---|
| DAFormer [11] | 0.00 | 66.19 | 45.29 |
| Ours (w./ CLIP [28]) | 31.31 | 68.73 | 56.91 |
| Ours (w./ ImageBind [7]) | **41.67** | **70.07** | **61.10** |

**Table 6: Comparison of our method on various inconsistent taxonomy settings.**

| | Coarse-to-fine | | Implicitly overlapping | |
|---|---|---|---|---|
| | $MIoU_{fine.}$ | $MIoU_{all}$ | $MIoU_{impl.}$ | $MIoU_{all}$ |
| DAFormer | 27.07 | 57.68 | 28.02 | 51.56 |
| Ours | **50.10** | **63.71** | **51.87** | **56.94** |

we explore the impacts of the parameter $\sigma_1$ on excluding noisy novel class proposals. We build our experiments on the DAFormer [11] framework. Table 4 illustrates a comparison of the performance under different $\sigma_1$ values when $\sigma_2 = 0.1$. It is evident that as $\sigma_1$ gradually increases, the predictive accuracy also shows an improvement, where its peak performance is achieved when $\sigma_1$ reaches 2. This occurs because a smaller $\sigma_1$ may result in challenging regions within shared classes being incorrectly relabeled as novel classes. In these cases, the number of trainable samples in shared classes is reduced and a significant amount of misleading information is introduced into the training of novel classes, hindering the robust learning of both shared and novel classes. Whereas, when the $\sigma_1$ becomes excessively large, for instance, when $\sigma_1$ is set to 4, a significant decline in performance is observed. This is due to the circumstance that, upon satisfying the specified threshold, the limited region is being reassigned with correct novel class labels, resulting in the underutilization of novel class proposals for pseudo labels.

**Influence of the parameter $\sigma_2$.** We also conduct experiments to analyze the sensitivity of our proposed method on the parameter $\sigma_2$ within the DAFormer [11] framework, evaluating its performance on the SYNTHIA → Cityscapes benchmark. We summarize the segmentation results with the different $\sigma_2$ in Table 4 when $\sigma_1 = 2$. Observably, the model achieves optimal segmentation performance when $\sigma_2$ is configured as 0.1. Specifically, setting $\sigma_2$ to 0.1 results in an MIoU value of 41.67 for the novel classes and an overall MIoU

value of 61.10 across all classes. However, when $\sigma_2$ is excessively high or low, the performance of our model experiences degradation. When the parameters are set too low, such as 0, challenging regions within the shared classes are incorrectly relabeled as novel classes, reducing the quality of trainable samples for novel classes. Larger values reduce the number of novel class proposals to be stored, thereby limiting pseudo-label augmentation and diminishing the enhancement effects on our segmentation model.

### 4.5 Feasibility on VLM-based implements

In this section, we assess the feasibility of our method under different VLM-based implementations. Here, we provide several alternative solutions with different VLMs to classify mask proposals for novel classes. Specifically, ImageBind and CLIP [28] are involved in the evaluation. Based on the aforementioned implementations, we retrain our model with the DAFormer framework[11] on the SYNTHIA → Cityscapes benchmark. As shown in Table 5, in addition to ImageBind, our method also displays positive effects with CLIP, where the mIoU accuracy of novel classes is 31.31 and that of all classes is 56.91. However, in comparison to CLIP, our model demonstrates superior performance on ImageBind, exhibiting a performance gain +10.36 for novel classes and +4.19 for all classes. This may be because ImageBind conducts contrastive pre-training on multiple modalities (i.e., audio and depth images), contributing to learning more robust visual-language connections of classes. This can be advantageous to obtain more precise mask proposals for novel classes. These results demonstrate that our method can be implemented with different VLMs.

### 4.6 Generalization on various taxonomy types

In TACS [8], three types of inconsistent taxonomic setups are studied, including the open, coarse-to-fine, and implicitly overlapping taxonomy settings. In Tables 1 and 2, we assess the effectiveness of our method under the open taxonomy setting in the *zero-labeled* target domain. Here, we further evaluate the adaptability and effectiveness of our DAFormer-based pipeline under other two distinct taxonomy settings. All experiments here strictly follow the settings in TACS, with the only variation being the zero-labeled target domain. Table 6 presents the results in the coarse-to-fine and implicitly overlapping taxonomy settings, where our method has a positive impact on both settings. The performance improvements demonstrate the generalizability of our proposed method across various taxonomy types.

## 5 CONCLUSION

In this paper, we present a generic self-training framework for addressing the CCDA problem in semantic segmentation. Our proposed label alignment and novel class resampling methods effectively rectify pseudo labels and enhance the supervision of novel classes, offering a promising solution for both cross-class and cross-domain challenges. In comprehensive experiments, we have shown that our method archives considerable performance improvements when embedded into UDA approaches across various settings, such as different benchmarks and different taxonomy types. Importantly, our method exhibits versatility to diverse VLM-based implements, enhancing its flexibility for real-world applications.

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
