# OpenReview forum: "Cross-Class Domain Adaptive Semantic Segmentation with Visual Language Models"
_acmmm.org/ACMMM/2024/Conference — MM2024 Poster_

### Official Review · Reviewer_KkUX · 2024-05-24

**Rating:** 4
**Confidence:** 1

**Summary:**

In addressing the emergence of novel categories in semantic segmentation domain adaptation, the paper leverages vision language model to facilitate zero-shot category prediction and employs confidence-based filtering to enhance the removal of unreliable predictions. Experimental results demonstrate that the proposed method exhibits performance improvements when integrated into various Unsupervised Domain Adaptation (UDA) semantic segmentation approaches. From a task perspective, semantic segmentation is a basic visual task, and domain adaptation is also a traditional visual problem. However, the paper starts from the annotation of semantic tags and uses a visual language model to bring new vitality to this traditional task. Although the semantic segmentation task cannot be called a multi-modal task, the Cross-Class Domain Adaptation method designed in the paper is consistent with the idea of ​​multi-modal research. Although I don’t know much about the field of semantic segmentation, I think the structure of the paper is reasonably arranged, the experiments are complete, and the logic is smooth. It is a qualified paper in the field of computer vision.

**Strengths:**

+ The paper's utilization of vision language model's robust generalization capabilities for annotating unknown categories is intuitive and sound.
+ Using visual language models to annotate visual data and solve purely visual tasks takes advantage of multi-modal research.
+ The manuscript's writing style is easily comprehensible and coherent.
+ The methods compared in the paper's experiments are all recent state-of-the-art (SOTA) approaches, and their effectiveness is demonstrated on two distinct datasets.

**Limitations:**

+ In the experiment from SYNTHIA to Cityscapes, the paper conducted experiments on DACS, DAFormer, and MIC, but in the experiment from SYNTHIA to IDD, only the experiment was conducted on DAFormer. Is there any reason for this?
+ Due to my limited familiarity with this specific domain, I may not be able to provide detailed technical feedback.
+ I will adjust my scoring based on the comments from other reviewers and rebuttals.

**Suitability:**

2

---

### Official Review · Reviewer_9VBR · 2024-05-24

**Rating:** 3
**Confidence:** 3

**Summary:**

The paper addresses the problem of cross-class domain adaptation (CCDA) in semantic segmentation, where the target domain contains both shared and novel classes not present in the source domain. The proposed solution leverages visual language models (VLMs) to relabel pseudo labels for novel classes and uses memory banks to augment pseudo labels with accurate VLM predictions. The method is evaluated on multiple benchmarks and outperforms state-of-the-art approaches.

**Strengths:**

1.The paper presents a novel approach that uses VLMs to address the CCDA problem in semantic segmentation.

2.The paper provide a detailed explanation of their proposed method and its underlying principles.

**Limitations:**

1.Word spelling problem: Line85: VFMs->VLMs.

2.In Figure 3, the terms "Student Net" and "Teacher Net" are mentioned. However, the paper does not provide a clear explanation of what "Teacher Net" specifically refers to, nor does it mention "Student Net" elsewhere in the text. This lack of clarity makes it difficult to understand the overall methodology.

3.The mention of the Genetic Segmentation Model in Figure 3 lacks clarity regarding the specific method employed in the text. Additionally, comparative experiments are absent.

4.The paper needs clarification on the size and design of the Memory Banks used. Please provide more details for a better understanding of the system architecture.

5.The paper discusses the use of pseudo-labels in model training. However, the paper lacks a detailed analysis of the proportion of pseudo-labels and their impact on model accuracy.

Suggestions:
1.Could you please clarify why you chose to use VLMs for the CCDA problem? What are the advantages of using VLMs over other methods?

2.Can you discuss any potential limitations or drawbacks of your proposed method?

**Suitability:**

2

---

### Official Review · Reviewer_KW6b · 2024-05-27

**Rating:** 4
**Confidence:** 2

**Summary:**

This paper proposes a Cross-Class Domain Adaptive (CCDA) framework for semantic segmentation, aiming to address the challenge of target domains containing both shared and novel classes that are either unlabeled or unseen in the source domain. This task is challenging because the absence of labels for novel classes hinders the accurate segmentation of both shared and novel classes in the source and target domains. To tackle this issue, the authors leverage the capabilities of Visual Language Models (VLMs), which can generate zero-shot predictions without requiring task-specific training examples.

Specifically, the main contributions of the paper include:

Label Alignment Method: A VLM-based label alignment method is proposed, which uses an uncertainty-driven threshold to relabel pseudo labels for novel classes, thereby enhancing the accuracy of segmentation.

Novel Class Resampling Method: A novel class resampling method is designed, employing an adaptively updated memory bank to effectively manage accurate VLM predictions and increase the sampling probability of novel classes, thus enhancing the supervision for novel classes.

Experimental Validation: The effectiveness and versatility of the proposed method are demonstrated through extensive experiments on various benchmarks, UDA frameworks, and VLM-based segmentation methods.

The paper also provides detailed descriptions of the proposed method's framework overview, problem statement, methodology, including specific steps for label alignment and novel class resampling, and showcases the performance of the method through experimental settings and results.

The significance of this work lies in providing an effective solution for handling common cross-class domain adaptation problems in the real world, especially in semantic segmentation tasks, where it can better manage the differences between the source and target domain classes.

**Strengths:**

Innovation: A novel Cross-Class Domain Adaptive (CCDA) method has been proposed, specifically addressing the challenge of the presence of novel classes in the target domain that are unmarked or unseen in the source domain, an issue that is less explored in existing literature.

Application of VLMs: The use of Visual Language Models (VLMs) is leveraged for zero-shot learning to tackle the problem of the lack of annotations for novel classes in the target domain. VLMs are capable of generating image-level predictions that match text descriptions, offering a potential solution for the segmentation of novel classes.

Label Alignment: Through the label alignment method, the paper introduces a strategy that utilizes VLMs to relabel pseudo labels, which helps to improve the model's segmentation accuracy for novel classes.

Novel Class Resampling: A novel class resampling mechanism has been designed, which dynamically manages the novel class proposals generated by VLMs through a memory bank and enhances the learning efficiency of novel classes with an adaptive updating strategy.

Uncertainty-Driven Threshold: The paper proposes an uncertainty-based adaptive threshold method to exclude noise in VLMs' predictions, thereby improving the accuracy of label alignment.

**Limitations:**

Dependency on VLMs' Performance: Since this method relies on Visual Language Models (VLMs) to generate pseudo labels for novel classes, the performance limitations of VLMs may affect the overall effectiveness of the framework.

Computational Resources: The computation of VLMs and the adaptive threshold may require substantial computational resources, which could limit the application of this method in environments with limited resources.

**Suitability:**

3

---

### Meta-Review · Area_Chair_MhMZ · 2024-07-01

**Recommendation:** Accept (Poster)
**Confidence:** 3

**Metareview:**

The AC goes through the paper, rebuttal and review comments. This paper got 3 borderline accept reviews. All reviewers acknowledge the motivation and novelty. Therefore, the AC thinks this paper can be accepted, but the authors should carefully improve the paper by leveraging the feedback.